# Effect of Cr, Mo, and V Elements on the Microstructure and Thermal Fatigue Properties of the Chromium Hot-Work Steels Processed by Selective Laser Melting

**Mei Wang** [1] , **Bo You** [2] **, Yan Wu** [3] **, Bo Liang** [1] **, Xianhui Gao** [4] **, Wei Li** [4],* **and Qingsong Wei** [4],*

[1] Rongcheng College, Harbin University of Science and Technology, Rongcheng 264300, China; wangmei@hust.edu.cn (M.W.); bliang0325@163.com (B.L.)
[2] School of Mechanical and Power Engineering, Harbin University of Science and Technology, Harbin 150080, China; youbo@hrbust.edu.cn
[3] Institute of Advanced Integration Technology, Shenzhen Institute of Advanced Technology, Chinese Academy of Sciences, Shenzhen 518055, China; y0wu0020@louisville.edu
[4] School of Materials Science and Technology, Huazhong University of Science and Technology, Wuhan 430074, China; gaoxianhui@hust.edu.cn
* Correspondence: lwliwei1989@163.com (W.L.); wqs_xn@hust.edu.cn (Q.W.); Tel.: +86-(0)-2787558155 (Q.W.)

**Abstract:** Thermal fatigue is the main failure mode for chromium hot-work steels. In this study, pre-alloyed chromium hot-work steel powders with three different Cr, Mo, and V addition levels (low content (LH), medium content (MH), and high content (HH)) were used for selective laser melting (SLM). The microstructure and thermal fatigue properties of these SLM-processed materials were investigated. After thermal fatigue tests, LH possessed the lowest hardness (approximately 573 $HV_5$) and longest crack length, MH possessed the highest hardness (approximately 688 $HV_5$) and HH (with the hardness of approximately 675 $HV_5$) possessed the shortest crack length. It can be concluded that the increase of V content in MH is the main reason for the refined grains which result in an enhanced hardness and thermal fatigue resistance compared to LH. The further increase of the Cr and Mo content in HH leads to the grain coarsening and hardness decreasing, which is supposed to degrade the thermal fatigue resistant properties according to the conventional theory. However, HH exhibited an enhanced thermal fatigue resistance compared to MH. That is because the higher stored energy in MH deteriorated its thermal fatigue resistance compared to HH.

**Keywords:** selective laser melting (SLM); chromium hot-work steels; alloying elements; high-temperature properties; thermal fatigue

## 1. Introduction

Chromium hot-work steels are commonly used in hot-work applications, i.e., stamping, hot rolling, hot forging, die casting, etc. During the forging process, the dies are repeatedly subjected to heat/cool cycles and mechanical loads which develop from the close contact between tools and hot workpieces [1,2]. These severe conditions generally can lead to surface damages, such as heat checking, erosion, and corrosion [3]. Heat checking originating from thermal fatigue (TF) is the most common failure of hot-work dies, and its process is significantly influenced by the non-uniform expansion caused by the thermal gradients from the surface to the center of the die [3–5]. As a result, hot-work dies can benefit from incorporating methods of increased and balanced heat transfer within dies by the careful controlling of surface temperature [1,6]. However, due to the constraints of conventional manufacturing, it is usually difficult to position the cooling or heating channels close to the surface of the cavity in a way that provides optimal heat transfer.

Selective laser melting (SLM) is an additive manufacturing technology that can fabricate high-density parts with complex geometries [7,8]. The advantages of SLM technology in mold applications include geometrical freedom, shortened design and product time,

reduction in process steps, customization, and material flexibility [2]. Increasingly, SLM is used to produce molds with conformal cooling channels. It is published that this type of mold can eliminate the defects caused by the poor cooling system of the molds [9], reduce cycle time, shrinkage, and porosity [10], and diminish surface defects as well as grain coarsening [11]. Therefore, the SLM-processed chromium hot-work dies exhibit great potential in the mold field.

Recently, extensive studies have been carried out to illustrate the relationship between the microstructure and the mechanical properties of hot-work steels fabricated by SLM. Huber et al. revealed that the as-built AISI H11 hot-work steels possessed a bainitic structure since the preheating temperature (approximately 380 °C) during SLM was higher than the martensitic transformation temperature [1]. What is more, a hardness of $642 \pm 9$ HV1 is achieved. The annealing treatment (550 °C for 2 h) resulted in an increase in the hardness ($678 \pm 8$ HV1) and tensile strength ($2148 \pm 16$ MPa) due to the decomposition of residual austenite or the precipitation of secondary carbides. Holzweissig et al. found that the metastable-retained austenite in the as-built H13 transformed into martensite after a tensile test [3]. Our previous work revealed that the microstructures of the as-built H13 transformed from the martensite and retained austenite mixtures to ferrite and carbides mixtures after tempering treatment [4]. The materials exhibited good stabilization of the hardness when treated below 550 °C, while a considerable hardness decrease occurred at 600 °C. The stability of the material was strongly dependent on the dissolution of the cell-like substructures and the stability of the martensite. However, as one of the most important indices, the TF properties of hot-work steels processed by SLM have been rarely investigated. Our previous work reported that the SLM-processed H13 exhibited shorter total crack length and higher hardness values after TF tests compared to the forged counterparts [9]. Microstructural investigations revealed that the increased amount of retained austenite, typical cell-like substructures, and refined grain size were the main reasons for the enhanced TF resistance properties of the SLM-processed H13. However, the effect of alloy elements on the TF properties of H13 was not considered. Nonetheless, in fact, alloying elements have a significant effect on the properties of SLM-processed parts. Harrison et al. increased the most potent solid solution strengthening elements in the Hastelloy X, and they found that minor variations in alloy elements (less Ni, Cr, and Mn contents, and more Co and W contents) led to a 65% reduction in crack density and an increase in tensile strength [10]. Li et al. applied V carbides to reinforce SLM-processed 316 L stainless steel and obtained a tensile strength (higher than 1400 MPa) more than twice as high as that of the material without V carbides [12]. Jia et al. added Mn and Sc into Al alloy and obtained a high strength Al alloy. They found that due to the rapid solidification of the SLM process, a high solid solution of Mn and a high density of nano-sized $Al_3Sc$ precipitates were obtained, which strengthened the material [13]. Apparently, the variation of alloying elements can change the properties of the SLM-processed materials.

Previous studies have shown that the hot-work steels were mainly composed of tempered martensite and alloy carbide precipitates, along with a high density of dislocations [14]. The tempering resistance and the high-temperature fatigue strength of this material can be improved by postponing the coarsening of fine carbides [14]. Therefore, it is particularly important to study the influence of carbide forming elements on the TF resistance behavior of SLM-processed chromium hot-work steels. Generally, Cr is usually added to increase resistance to corrosion, the hardenability, and the high-temperature strength of alloyed steels [15]. During tempering, Cr can produce secondary hardening and hinder oxidation kinetics. Furthermore, the aggregation of carbides can be hindered by substituting Cr for some of the iron in cementite [16]. Mo (normally 0.5% to 2%) can improve the toughness and resistance to the cracking of castings [15]. It can also produce secondary hardening during tempering and enhance the high-temperature creep strength of low-alloy steels. V can hinder the growth of carbides and improve resistance to tempering. It improves resistance to hydrogen attack but may promote the occurrence of hot (reheat) cracking. In addition, it contributes to grain refinement and the enhancement of

yield strength in steels, but has a negative effect on elongation [17]. In conclusion, Cr, Mo, and V elements play an important role in oxidation resistance, hardness, strength, and the toughness of steels. Since it has been illustrated that these properties could significantly determine the TF properties of steels [18–20], it is believed that the optimization of the content of Cr, Mo, and V can improve the TF properties of hot-work dies.

In the present study, the effect of carbide forming elements (Cr, Mo, and V) on the microstructure and TF properties of the chromium hot-work steel fabricated by SLM was systematically investigated. The TF resistance mechanism of SLM-processed H13 was revealed. We expect that the current research can be applied to the development of high-performance chromium hot-work steels, especially for additive manufacturing processes.

## 2. Experimental Procedures

### 2.1. Materials

The gas atomized pre-alloyed powders of chromium hot-work steel with three different amounts of Cr, Mo, and V elements (including the low content (LH), the medium content (MH), and the high content (HH)) were used in this study. As shown in Figure 1, these powders exhibited a nearly spherical profile with some unexpected small planar balls (indicated with arrows). The average powder sizes were approximately 20.12 μm, 27.07 μm, and 24.57 μm for LH, MH, and HH, respectively. The detailed chemical compositions of the three materials are given in Table 1. Chemical compositions were determined by the methods that were used in reference [4].

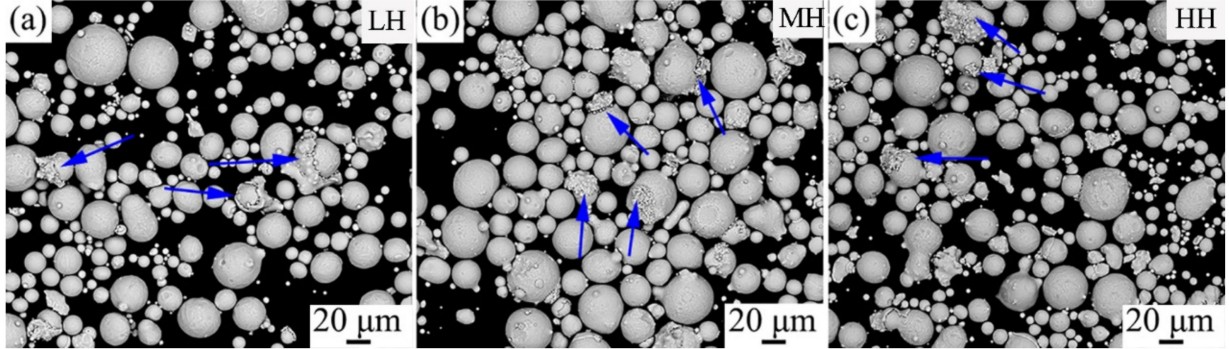

**Figure 1.** SEM images of the pre-alloyed powders of (**a**) LH, (**b**) MH, and (**c**) HH.

**Table 1.** The chemical composition of the pre-alloyed powders. (wt %).

| Material Type | C | Cr | Mo | Si | V | Mn | Fe |
|---|---|---|---|---|---|---|---|
| LH | 0.34 | 4.77 | 0.89 | 0.94 | 0.54 | 0.48 | Bal. |
| MH | 0.37 | 5.05 | 1.32 | 1.11 | 1.07 | 0.49 | Bal. |
| HH | 0.36 | 5.72 | 1.47 | 1.08 | 1.07 | 0.68 | Bal. |
| Tolerance (±) | 0.005 | 0.035 | 0.024 | 0.020 | 0.020 | 0.015 | |

### 2.2. SLM Process

The bulk materials were fabricated from the aforementioned powders by an SLM machine (SLM 125 HL, SLM Solutions GmbH, Lübeck, Germany). The focused laser beam diameter is 80 μm with a Gaussian distribution of power density. The optimized processing parameters for high-density parts were as follows: laser power, 280 W; scanning speed, 980 mm/s; hatching spacing, 120 μm; and layer thickness, 40 μm. A parallel scanning line rotated by 67° relative to the previous layer was chosen. During the SLM process, a pure argon atmosphere was maintained and the platform temperature was approximately 90–93 °C.

### 2.3. Experiments and Methods

Figure 2a shows the bulk materials after SLM. The round bars were used for tensile tests. The rectangular blocks were used for TF tests. No cracks have been observed in these

materials. Figure 2b shows the profile of specimens for TF tests. The obtained hardness values were $528 \pm 8$ HV$_5$, $597 \pm 12$ HV$_5$, and $592 \pm 9$ HV$_5$ for LH, MH, and HH, respectively. The densities were $7.759 \pm 0.0012$ g/cm$^3$, $7.754 \pm 0.0037$ g/cm$^3$, and $7.778 \pm 0.0010$ g/cm$^3$ for LH, MH, and HH, respectively, according to the Archimedes method.

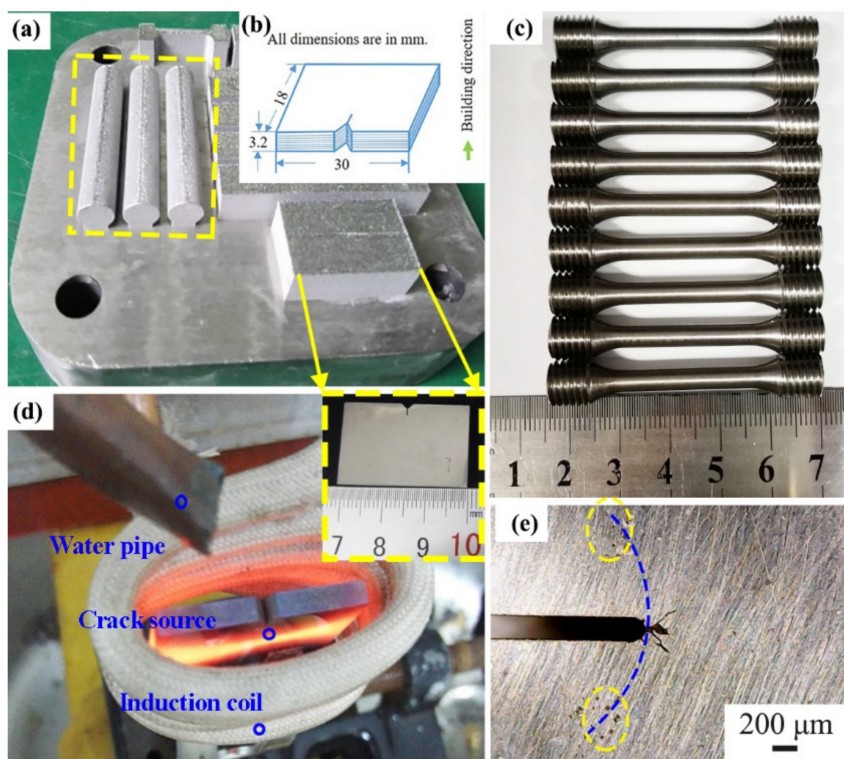

**Figure 2.** (**a**) SLM-processed bulk materials. (**b**) Profile of the specimen for TF tests. (**c**) Specimens for tensile tests. (**d**) Photo of the apparatus for TF test, adapted from ref. [9]. (**e**) Photo indicated the location where the hardness indents were performed.

It has been published that materials with higher yield strength generally exhibit shorter crack length [18]. Therefore, in order to study the TF properties, tensile tests and TF tests should be carried out. Figure 2c shows the tensile test specimens, which were CNC machined (wire-cut EDM machine C400iC, Fanuc, Yamanashi, Japan) from the bulk materials. Test pieces of round cross-sections, with an original gauge length of 23 mm and an original diameter of 4 mm, were machined along the horizontally (perpendicular to the build direction) built samples. High-temperature tensile tests were performed by a precision universal tester machine (AG-IC 100 kN, Shimadzu Corp., Kyoto, Japan), with a strain rate of 1 mm/min. Before the tensile tests, all the specimens were heated to 600 °C and held for 5 min according to ISO 783. For each grade of the materials, three specimens were used for testing. The loading direction for the tensile tests was perpendicular to the building direction. The yield strength was determined by the stress corresponding to the intersection of the stress–strain curve offset by a specified strain of 0.2%. In the present study, a TF test was carried out by a self-made machine (Figure 2d) with a 9 s test cycle (3 s for heating to 650 °C using an induction coil and 6 s for cooling to 30 °C using tap water, according to reference [21]). Specimens for TF tests (3 pieces of test samples for each grade) were prepared by low-speed wire electrical discharge machining (the profile of specimens is shown in Figure 2b). TF tests were interrupted every 500 cycles, and then the crack length and the hardness values were measured and studied. The crack was detected by an optical microscope, and the longest crack length was recorded. Hardness was measured by a Vickers hardness tester (432 SVD, Wolpert Wilson Instrument, Wuxi, China) with a load of 5 kg over 15 s. All hardness tests were conducted on the horizontal surface (perpendicular to the build direction). Figure 2e shows the location where the hardness

indents were performed (the dashed line indicated the 650 °C temperature area during TF tests). For each grade of the material, three specimens were tested to calculate the mean hardness values and the standard deviation. More information about the preparation of the specimens and the TF tests can be referred to [9].

Phase compositions of the three as-built materials were investigated by using an X-ray diffraction instrument (XRD-7000s, Shimadzu Corp., Kyoto, Japan) with Cu Kα radiation at 40 kV and 40 mA. In this study, all the three materials should form laths during the martensite transformation because their carbon compositions were all in the range of 0–0.6 wt % [22]. Laths consist of fine units hierarchically arranged in substructures within prior–austenite grains and usually exhibit packets and blocks of individual laths. It is known that the strength and toughness of conventional martensitic steels are strongly dependent on the size, shape, and arrangement of packets, blocks, and laths [23]. Therefore, investigations on the crystallographic features were carried out to reveal the relationship between the microstructure and the mechanical properties of SLM-processed chromium hot-work steels. The microstructure of the as-built specimens was observed by scanning electron microscopy (SEM, Siron 200, FEI Company, Eindhoven, Holland). EBSD investigations (EDAX Inc., Mahwah, NJ, USA) were carried out on the as-built specimens to study the crystal orientations, phase distributions, and grain boundary misorientations. The EBSD data were collected by an EDAX system and analyzed by a TSL OIM analysis software (version 6, EDAX Inc., Mahwah, NJ, USA). EBSD measurements were carried out on the surface perpendicular to the build direction. Furthermore, the location was in the middle of two laser tracks. The prior austenite grains, after TF tests, were observed by a super-high magnification lens zoom 3D microscope (VHX-1000C, KEYENCE, Osaka, Japan). Specimens for SEM and optical microscope investigations were etched with corrosive liquid (30 mL of saturated picric acid with 3 drops of hydrochloric acid and 1 drop of detergent) at 60–80 °C for 20–60 s. Specimens for EBSD investigations were etched with A2 reagent (vol 10% perchlorate, vol 90% ethanol) using an electrolytic polish machine (LectroPol-5, Struer Als, Ballerup, Denmark) at 20 V for 200–300 s.

## 3. Results

### 3.1. Phase Analysis

During the equilibrium solidification process of 5% Cr hot-works steel, carbides precipitation in the α-iron matrix should be developed according to the Fe-C (5% Cr, the blue arrow) binary phase diagram of Figure 3a. However, carbides were not observed in these three materials. On the contrary, these carbides form elements aggregated as cell-like substructures because of the rapid cooling speed of the SLM process [4,24]. Figure 3b shows that all the as-built SLM-processed LH, MH, and HH materials exhibited the same phase compositions of α-iron and γ-iron. Some austenite is retained for the rapid solidification of the SLM process. Since the metallic alloying elements contain atoms larger than the iron and these atoms enter the matrix as substitutional atoms, an increase in lattice plane distance should take place with the increase in the content of alloying elements. As a result, the decrease of the $2\theta$ value should be detected according to Bragg's law [25],

$$2\ d\ sin\theta = n\ \lambda, \ (n = 1, 2, 3 \ldots ), \tag{1}$$

where $d$ is the lattice plane space, $2\theta$ is the diffraction angle, $\lambda$ is the wavelength, and $n$ is a diffraction order. However, Figure 3c reveals that the $2\theta$ degree of the (110) peak of α-iron increased with the increase in the content of alloying elements. These contrary findings in this paper may be caused by the following reasons. As revealed by the EBSD phase distribution images of Figure 3d–f, the fraction of the austenite increased from 5.1% to 16.8% while increasing the content of alloying elements. It is known that carbon atoms are sufficiently small and enter the α-iron lattices as interstitial solute atoms [26], and the solubility of carbon in γ-iron is much higher than that in α-iron. Since the carbon contents of the three materials were very close and presented in solid solution [27], the increase in the fraction of γ-iron led to a decrease in the solid solution of carbon in α-iron and fewer

amounts of C remained in α-iron, which resulted in less lattice distortion and the increase of the *2θ* value. In conclusion, although substitutional atoms of metallic alloying elements increased with the increase in the content of alloying elements, fewer amounts of C were left in α-iron, resulting in the less lattice distortion and the increase of the *2θ* value.

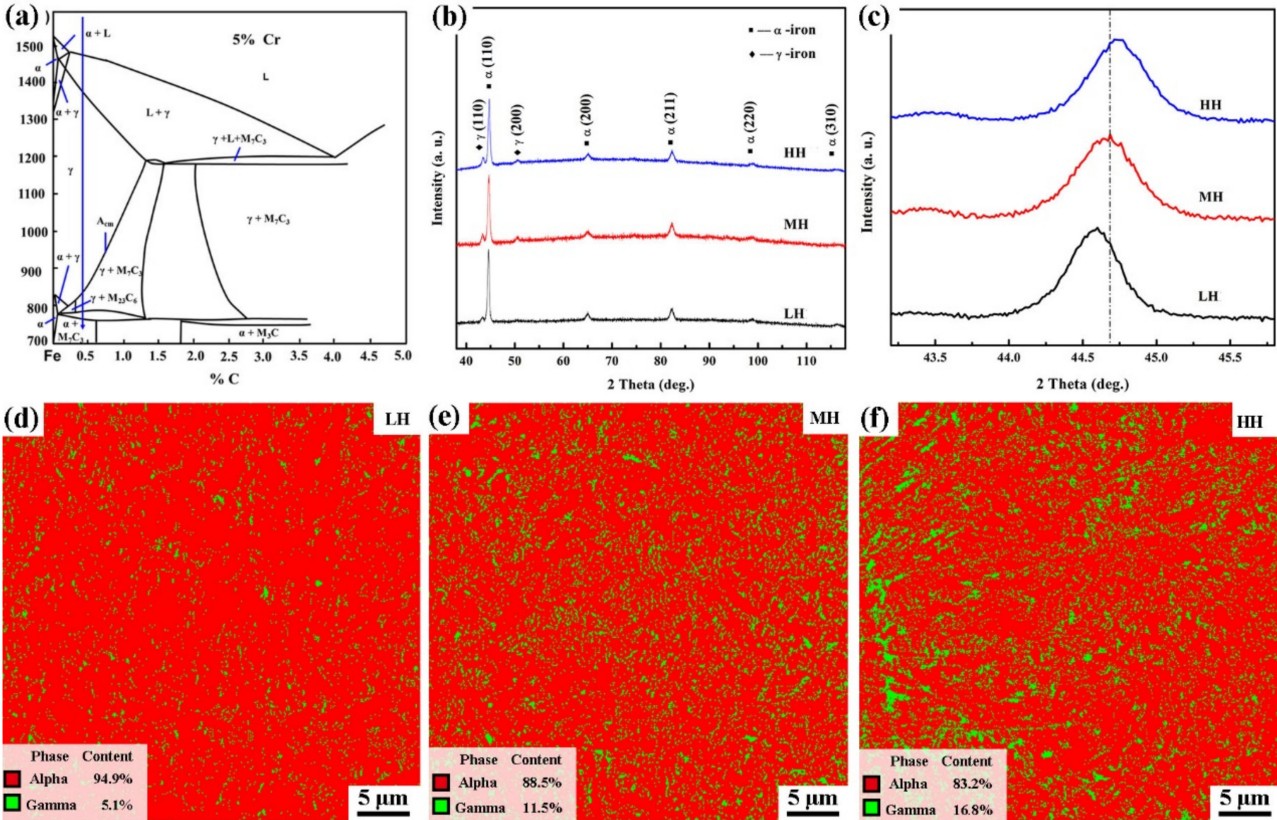

**Figure 3.** (**a**) The Fe-C (5% Cr) binary phase diagram, adapted from ref. [4]. (**b**) The XRD spectra of the as-built SLM-processed LH, MH, and HH. (**c**) The magnified image of the major α-iron peaks (110) of (**b**). (**d**–**f**) Phase distribution images of the as-built SLM-processed LH, MH, and HH.

### *3.2. Crystallographic Features*

As shown in Figure 4a, SLM-processed LH exhibited the typical cell-like substructures distributed homogeneously in the matrix. However, they were not grains [28]. As the previous work illustrated, during the solidification process of SLM, fine columnar grains of prior austenite (<100> crystallographic direction) grew from the melt pool boundaries due to epitaxial grain growth [4,29]. In the meantime, cell-like substructures grew epitaxial along the fusion line due to the slow kinetics of homogeneous alloying of large atoms of heavier elements [4]. Martensitic grain developed when the temperature cooled down below the martensitic transformation temperature. The martensite transformation occurred inside the austenite grain, with some portion of residual austenite remaining.

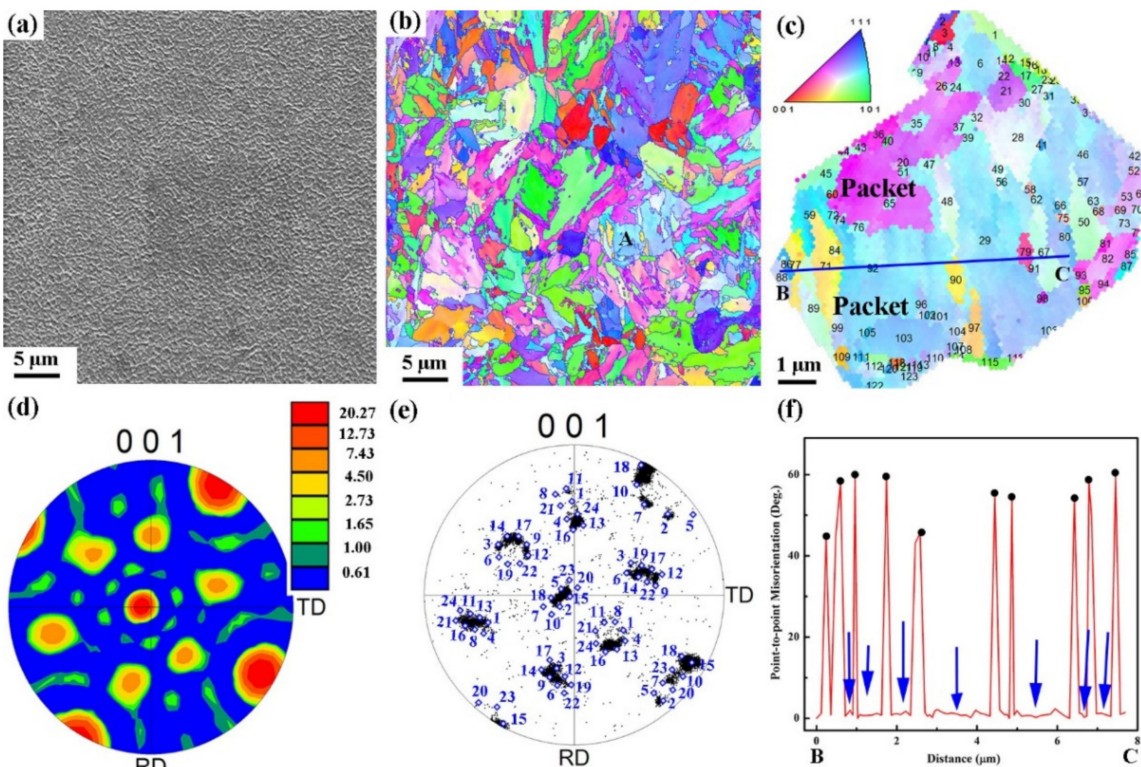

**Figure 4.** (**a**) SEM image of the as-built SLM-processed LH. (**b**) Corresponding inverse pole figure (IPF) figure of (**a**). (**c**) The cropped region A from (**b**). (**d**) The corresponding ODF figure of (**c**). (**e**) {001} pole figure shows the orientation of the martensite crystals within the prior austenite corresponds to (**c**). The symbols and numbers were corresponding to the variant numbers described in reference [23]. (**f**) Distribution of misorientation values along BC line.

In this study, EBSD results showed that the crystallographic features of all three materials were almost the same. Therefore, the microstructures of LH were used in this section to reveal the crystallographic features of the as-built SLM-processed chromium hot-work steels. Figure 4b is the corresponding inverse pole figure (IPF) of Figure 4a. It reveals that LH possessed the microstructure of martensite. EBSD results revealed that LH possessed a {001} texture and the texture intensity was calculated to be 3.5. The texture intensities were 10.5 and 4.0 for MH and HH, respectively. Therefore, the three materials possessed an anisotropic feature, as the texture intensity values were higher than 1.

Since the martensitic grains were transformed from the parent austenite, a further investigation of the orientation relationships between the martensite and its parent austenite in one single parent austenite grain was carried out (Figure 4c, the cropped region A from Figure 4b). Figure 4d reveals that the texture index of martensite that transformed from the same prior grain was approximately 20.27. The ideal orientations of the 24 variants were rotated to coincide with the actual orientation of the measured martensite, which was superimposed with the diamond symbols in the {001} pole figure, Figure 4e. Generally, when one parent austenite transforms to martensite, three different orientation relationships can be observed between the parent austenite and the inherited martensite in steels, including Kurdjumov–Sachs (K–S), Nishiyama–Wassermann (N–W), and Greninger–Troiano (G–T) [23,30]. It can be concluded from Figure 4e that the concentrations near the ideal orientations based on the K–S orientation were obvious [23]. Figure 4c also identifies that the SLM-processed LH exhibited similar microstructural features as that of the conventional martensitic steels. In other words, one parent austenite grain consists of several packets which are divided into parallel blocks and further subdivided into laths. As shown in Figure 4c, packets were marked with different colors. To understand the orientation change between and within the blocks or laths, the point-to-point misorientation analysis along

line BC was conducted. As shown in Figure 4f, block boundaries (marked with black points) had a misorientation of 40°–60°, whereas the laths boundaries were less than 10.53° (marked with blue arrows) [23]. Figure 4f also reveals that there were about 10 blocks across 8 μm in length, and each block was composed of several laths (indicated by the blue arrows).

### 3.3. Distributions of Grain Size

Lath martensite possesses boundaries, i.e., lath boundary, sub-block boundary, block boundary, packet boundary, and prior austenite grain boundary. Among these boundaries, the block boundary is the most effective grain boundary for the strength and the impact property of lath martensite attributed to its large misorientation [31]. Since subblocks should have misorientations less than 10.53° [23], misorientations larger than 10.5° were used in the present study to identify the block boundaries and to calculate the block size. As shown in Figure 5a–c, the average block sizes were approximately 2.56 μm, 1.43 μm, and 1.72 μm for LH, MH, and HH, respectively. Figure 5d–f shows that most of the blocks in these specimens were smaller than 4 μm, and the area fractions of the blocks larger than 2 μm were 49.6%, 24.7%, and 29.7% for LH, MH, and HH, respectively.

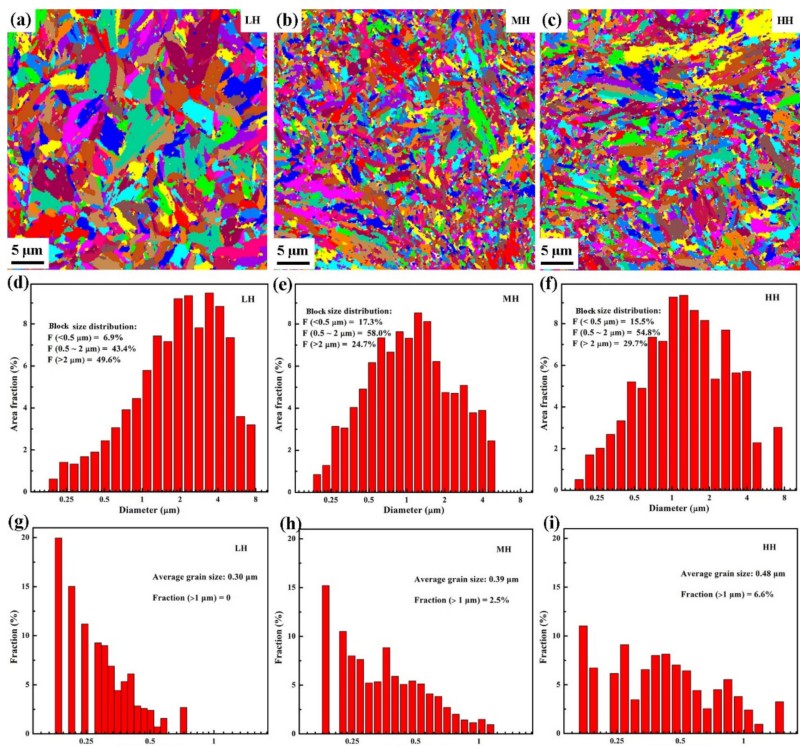

**Figure 5.** (**a**–**c**) Martensitic block maps. (**d**–**f**) The corresponding block size distributions. (**g**–**i**) The grain size distributions of retained austenite of SLM-processed LH, MH, and HH.

The grain size of the retained austenite has been investigated, since it determines the stability of the materials when tensile stress is applied [32]. It is published that austenite smaller than 0.01 μm is too stable to undergo the strain-induced transformation, while austenite larger than 1 μm is unstable and transforms quickly to martensite at small strains [32]. As shown in Figure 5g–i, the calculated average grain sizes of austenite were 0.30 μm, 0.39 μm, and 0.48 μm and the area fractions of the grains larger than 1 μm were approximately 0%, 2.5%, and 6.6% for LH, MH, and HH, respectively. The increased area fraction of austenite should result in the unstable and easy transformation of the material. On the whole, the grain size decreased first, then increased with the increase in the content of the alloying elements, whereas the grain size and the fraction of retained austenite increased all through.

*3.4. Coarse Laths*

Since laths with a misorientation that is smaller than 5° contain a high dislocation density [22,30], the Kernel average misorientations (KAMs) with the $\theta \leq 5°$ are generally used to determine the dislocations density [30,33]. KAM calculates the average misorientation between a data point and all of its neighbors, and the mean value of the KAMs is obtained by dislocation density ($\rho$) calculations, according to [33]:

$$\rho = 2\frac{v}{ub} \tag{2}$$

where $u$ is the unit length and $b$ is the Burgers vector, and $v$ is the mean value of the KAM (rad). According to Equation (2), the dislocation density ($\rho$) increases with the increase of the mean value of KAM. In the present study, KAMs were retrieved directly from EBSD data. Figure 6a shows that the misorientation angles were around 3.5°, 44.5°, and 60° for all three materials. For low angle boundaries, angles of all the three materials were typically in the range of 0.5–2.5°, and the peak value increased, then decreased with the increasing content of alloying elements (Figure 6b). This indicates that MH possessed the largest mean KAM value among the three materials, according to reference [33]. The EBSD results also proved this; the average KAM values were 1.448, 2.318, and 2.023 for LH, MH, and HH, respectively. Therefore, MH possessed the highest density of dislocation, and LH possessed the lowest according to Equation (2). Meanwhile, LH showed the largest portion of coarse laths (marked with arrows in Figure 6c–e) regions among the three materials. These coarse laths were characterized as less dislocated, significantly extended in size, and highly auto tempered. They were softer than the surroundings because they developed earliest in the course of martensitic transformation [34,35]. It was found that LH possessed the largest portion of coarse laths and the highest mean value of block size, while MH possessed the least portion of coarse laths and the lowest mean value of block size. Therefore, it can be concluded that the increase of the coarse laths happened along with the coarsening of block size.

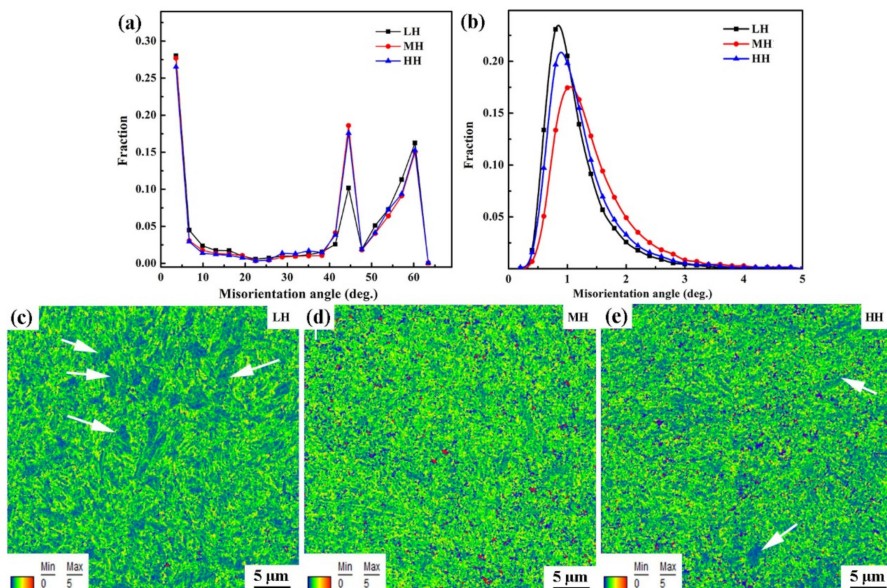

**Figure 6.** (**a**) Misorientation distributions of the as-built SLM-processed LH, MH, and HH. (**b**) The corresponding low misorientation angle distributions of (**a**). (**c**–**e**) The corresponding KAMs maps of (**b**).

*3.5. Tensile Properties*

The high-temperature (600 °C) tensile tests were conducted, and the summarized tensile properties are presented in Table 2. It was found that LH possessed a yield strength (YS) of 1090 ± 31 MPa, an ultimate tensile strength (UTS) of 1218 ± 18 MPa, and a total

elongation of 7.7 ± 0.3 %. The YS increased by approximately 94 MPa, then decreased by approximately 42 MPa, and the UTS increased by approximately 63 MPa, then decreased by 23 MPa with the increase in the content of alloying elements. Whereas, the elongation to failures decreased by approximately 0.6% and then increased by 1.5% with the increase in the content of alloying elements. Therefore, MH possessed the highest strength while possessing the least elongation among the three materials. Figure 7a–c reveals the fracture surface of the three materials after tensile tests. All three materials exhibited the ductile fracture feature with a density of dimples (Figure 7d–f).

**Table 2.** The summarized tensile mechanical properties at 600 °C.

| Material Type | Yield Strength (YS), MPa | Ultimate Tensile Strength (UTS), MPa | Total Elongation, % |
|---|---|---|---|
| LH | 1090 ± 31 | 1218 ± 18 | 7.7 ± 0.3 |
| MH | 1184 ± 26 | 1281 ± 19 | 7.1 ± 0.3 |
| HH | 1142 ± 10 | 1258 ± 22 | 8.6 ± 0.2 |

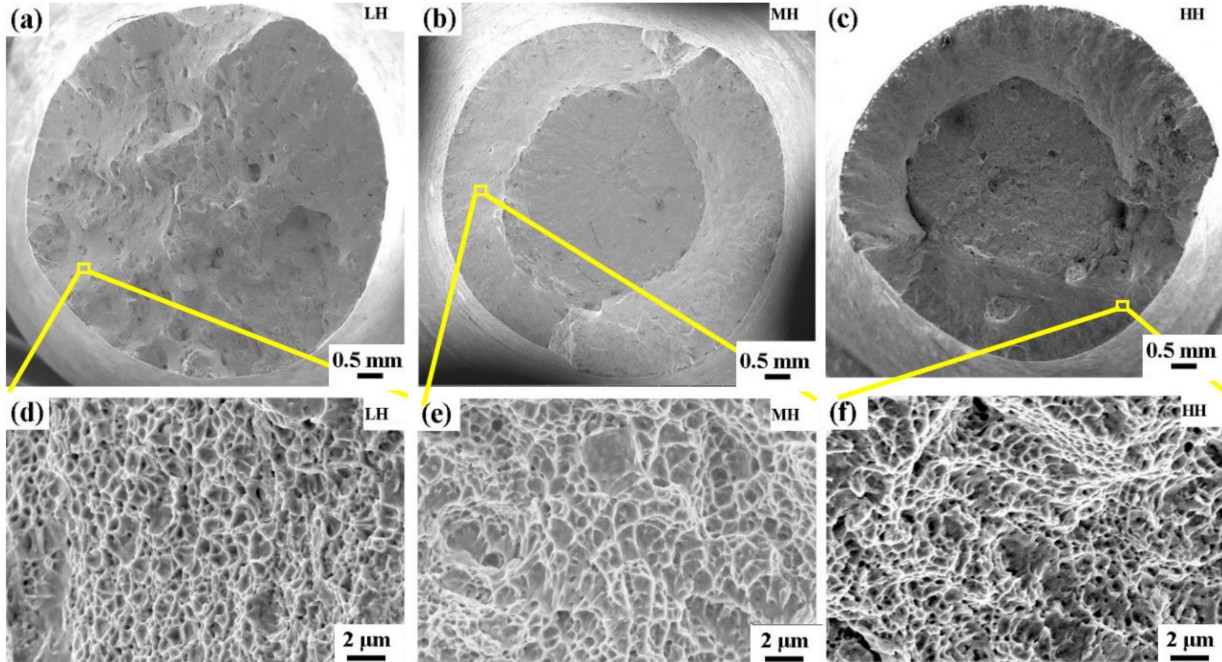

**Figure 7.** SEM images of the fracture surface of the SLM-processed LH (**a**) and (**d**), MH (**b**) and (**e**), and HH (**c**) and (**f**) parts.

Generally, the tensile properties of steels largely depend on the following aspects: grain refinement, solid solution strengthening, lath/plate strengthening, precipitation hardening, and retained austenite [22,32]. As revealed in Section 3.1, the fraction of retained austenite increased and the lattice distortion decreased with the increase in the content of the alloying element. Therefore, MH should exhibit a lower strength value and better ductility compared to LH, since MH possessed a larger fraction of retained austenite and less lattice distortion. However, grain refinement leads to the enhancement of the strength in MH. The high KAM value in MH led to a high level of stress concentration in the material [36], which may facilitate the propagation of cracks from the surface [37]. In addition, the presence of thinner laths (or low content of coarse laths) in MH led to inferior plasticity in the material compared to those materials with high content of coarse laths. As a result, the increase in strength and the decrease in ductility took place in MH. The decrease in strength and the increase in ductility revealed in HH should be attributed to the coarsening of grains and the increasing fraction of retained austenite in the material. Therefore, MH showed the highest strength while possessing the least elongation among the three materials.

### 3.6. Thermal Fatigue Properties

Figure 8 depicts the crack length evolution of the as-built SLM-processed materials during TF tests. Results show that LH possessed the shortest crack length after 250 cycles of TF tests, however, its crack propagation rate was faster than that of MH and HH. LH exhibited a crack length longer than the other two materials in the remaining fatigue cycles. HH exhibited the shortest crack length during the remaining TF tests. Since a few studies indicated that TF resistance properties could be improved with an increase in hardness [20,38], therefore, hardness after TF tests was investigated. Figure 9 depicts the microhardness evolution curves of the three materials after different cycles of TF. All the three materials exhibited secondary hardening behavior during TF tests, with the hardness increased by approximately 52 $HV_5$, 95 $HV_5$, and 74 $HV_5$ for LH, MH, and HH, respectively, after 1000 cycles. The hardness values were 573 ± 9 $HV_5$, 688 ± 11 $HV_5$, and 675 ± 10 $HV_5$ for LH, MH, and HH, respectively, after TF tests. No obvious softening behavior was detected in the three materials after TF tests. The secondary hardening behavior and the softening resistance behavior of the three materials during TF tests should be attributed to the solid solution strengthening of heavy elements and the precipitation of tiny carbides when treated at high temperatures [4]. It is known that softening (hardness loss because of the long-term operation at elevated temperature) would lead to the deterioration of TF resistance properties in hot-work dies. Therefore, the fantastic softening resistance properties of the three materials should lead to good TF resistance properties in the three materials.

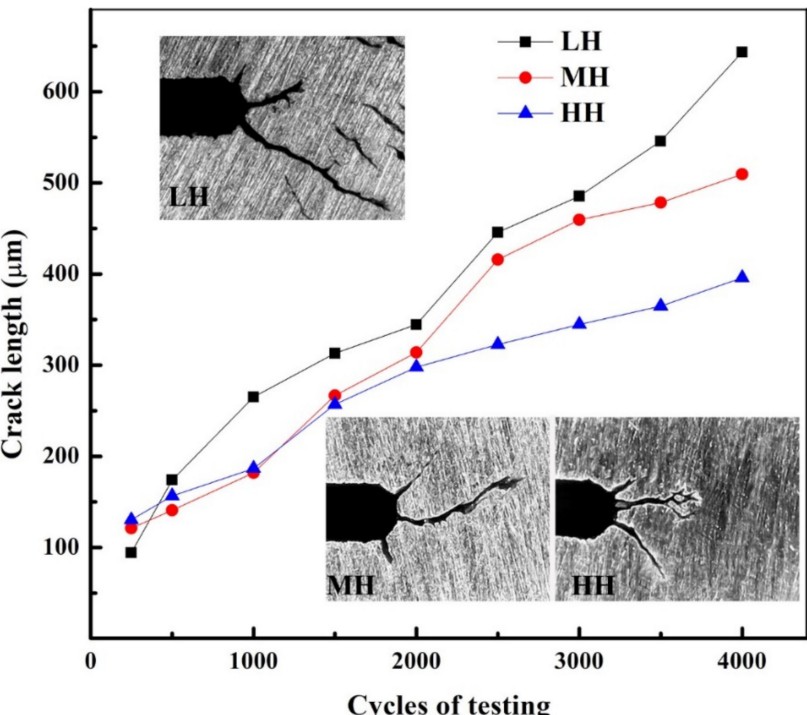

**Figure 8.** Crack propagation curves and crack images of the SLM-processed LH, MH, and HH during TF tests.

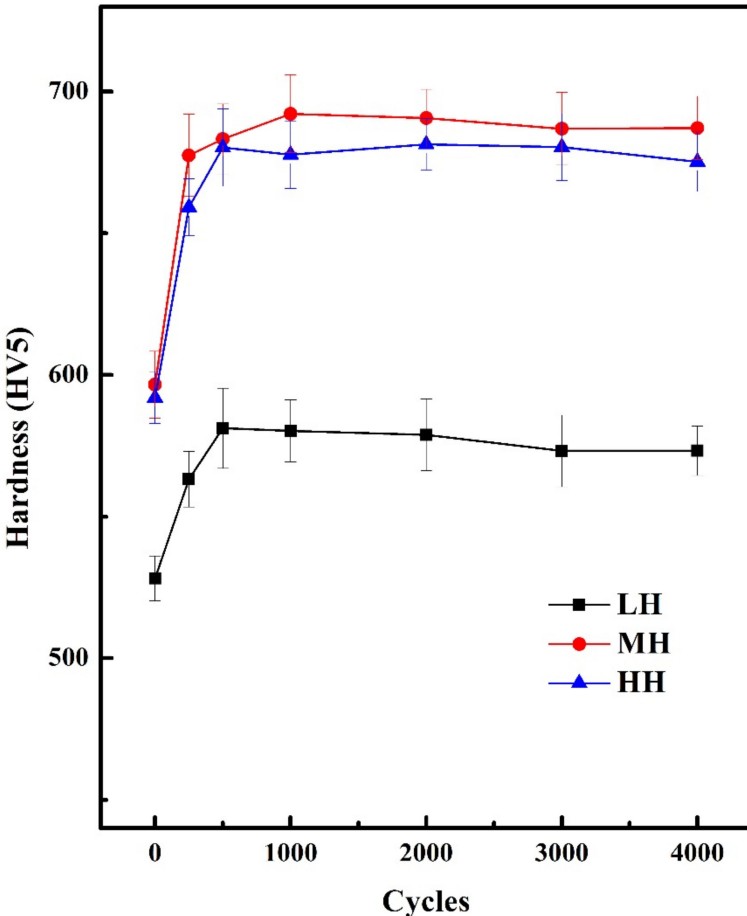

**Figure 9.** Microhardness after different numbers of thermal fatigue testing cycles (650 °C/30 °C). Curve of MH is cited from reference, adapted from ref. [9].

The reasons for the three materials exhibiting different original hardness were as follows. It is known that the yield stress of steels can be determined according to the Hall–Petch relationship [25,39]:

$$\sigma = \sigma_0 + k_y d^{-\frac{1}{2}} \tag{3}$$

where $\sigma$ is the yield stress, $d$ is the average grain diameter, $\sigma_0$ is the materials constant for the starting stress for dislocation movement, and $k_y$ is a constant for a particular material. The Vickers hardness $H_v$ of martensitic steels can be defined in terms of the yield stress $\sigma$ as the following equation [22]:

$$H_v = 0.4(\sigma + 110) \tag{4}$$

Equations (3) and (4) suggest that the yield stress and hardness increase with the decrease in grain size. Thus, LH exhibited the lowest yield stress and hardness as it possessed the largest block size among the three materials.

## 4. Discussion

*4.1. Effect of Alloying Elements on the Microstructures of the Chromium Hot-Work Steels Processed by SLM*

Generally, alloying elements have a significant influence on phase composition, grain size, and microstructural morphology of steels.

(1)  Retained austenite. Ordinarily, the decrease in the Ms (martensite transformation start) temperature would lead to an increase of austenite content [11]. The Ms temperature can be calculated as follows [40]:

$$\text{Ms } (^{\circ}\text{C}) = 521 - 353(\%\text{C}) - 225(\%\text{Si}) - 24.3(\%\text{Mn}) - 27.4(\%\text{Ni}) - 17.7(\%\text{Cr}) - 25.8(\%\text{Mo}), \tag{5}$$

According to Equation (5), it can be concluded that the increase of Cr, Mo, and Mn additions could lead to a decrease in Ms temperature and a further increase in the content of retained austenite [11]. Therefore, in this study, the area fraction of retained austenite increased with the increase in the content of Cr and Mo contents.

(2)  Block size. As presented earlier, the block size of MH decreased by approximately 1.13 μm (approximately 44.14% of the block size of LH) with the increase of Cr, Mo, and V contents. While the block size increased by approximately 0.29 μm with a further increase of Cr and Mo contents. Obviously, grain coarsening occurred with the increase of Cr and Mo in HH. Therefore, the increase in V contents should be the main reason for grain refinement in MH. This is coincident with the published theory that V could be added as a weak grain-refining element in steels [15].

(3)  Coarse laths. It is reported that coarse laths are softer than the surroundings since they experienced an auto temper process as they form earliest in the course of martensitic transformation [35]. According to the results of Sections 3.3 and 3.4, we found that LH possessed the largest block size and the largest portion of coarse laths among the three materials, whereas MH possessed the smallest block size and the least portion of coarse laths. Consequently, it can be supposed that the content of coarse laths increased with the increase in block size.

*4.2. Effect of the Microstructures and Mechanical Properties on the Thermal Fatigue Properties of the Chromium Hot-Work Steels Processed by SLM*

TF cracking is the dominant damage mechanism of chromium hot-work steel dies, which is facilitated by the stress-strain-temperature-time loadings and the environmental effects [18]. When dies are repeatedly subjected to heat/cool cycles and mechanical loads, the steep temperature gradient produces different expansion and contractions in the dies, and as a result, thermal stresses are induced. When the stress suppresses the material's yield strength, fine cracks initiate on the cavity surface and grow larger during the treatment, which finally leads to failure of the dies [15]. It is known that the low yield strength of materials would lead to an increase in crack length [18] and the increment in hardness would lead to a decrease in TF crack growth rate [20]. This can be illustrated by Equation (6). The level of the crack tip stress-concentration is estimated by $K_{IC}$ and is expressed as the following equation [41],

$$K_{IC} = \left[ c \left( \pi \frac{sa_e}{a+s} \right)^{\frac{1}{2}} \right] \left( \sigma_{0y} + k_y d^{-\frac{1}{2}} \right), \tag{6}$$

where $K_{IC}$ is the fracture toughness of the material, $c$ is a numerical constant, $\sigma_{0y}$ is the lattice frictional stress, $k_y$ are experimental constants, $\frac{sa_e}{a+s} \approx s$, where $s$ is the width of the crack tip, and $d$ is the grain's diameter. Apparently, $K_{IC}$ increases with the increase of $\sigma_{0y}$ or the decrease in grain size. Therefore, MH exhibited enhanced TF properties compared to LH should be attributed to the increased strength and decreased grain size of the material. However, it is abnormal to find that MH revealed inferior TF properties compared to HH, although it possessed higher strength and hardness than HH. This phenomenon is not in line with those published theories [20]. Therefore, the relationship between the microstructure and the resulted TF properties should be discussed in detail further.

(1)  Retained austenite. It is well accepted that retained austenite facilitates the toughness and ductility of steels [22]. The pileup occurs when dislocations move to the grain boundaries, which leads to stress concentration and crack initiation at the head of the pileup group. It was found that the moving resistance of the dislocations can be reduced by the coherent or semi-coherent interfaces (between the retained austenite and martensite matrix), and as a result, softens the materials and decreases the crack

initiation and propagation rate [42]. Furthermore, the transformation of retained austenite to martensite can develop the compressive residual stress in the material, which can suppress crack initiation and growth [43]. However, retained austenite only functions at the early stage of TF tests since it decreases largely because of the heating and cooling process.

(2)   Grain size. It is known that the strength and toughness of martensitic steels can be improved as the grain size is refined [31]. This is because the high-angle-grain boundaries (boundaries of prior austenite, packets, and blocks) strengthened by dislocation pileup possess the ability to impede the propagation of cleavage crack [31]. Since the temperature for the TF test was much higher than for the Ms temperature, the fracture toughness should be estimated by the grain size of the prior austenite. As is shown in Figure 10, LH, HH, and MH revealed their prior austenite grain sizes in descending order.

(3)   Coarse laths. Coarse laths represent soft spots in martensite and therefore they plastically yield first, leading to the early plasticity activity and becoming prone to bulk or interface plasticity. The coarse laths interiors also show the pronounced slip activity, whereas less slip activity is observed in the neighboring thin laths [35].

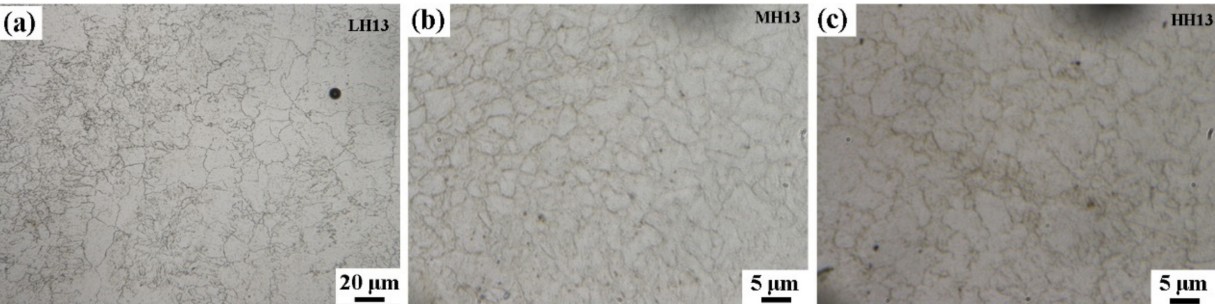

**Figure 10.** Grain maps given by the optical microscope of (**a**) LH, (**b**) MH, and (**c**) HH after thermal fatigue tests.

As stated before, with the increase in the content of alloying elements, the area fraction of retained austenite increased all throughout, the prior austenite size and the fraction of coarse laths decreased first then increased, and the strength and the hardness increased first then decreased. Therefore, LH should exhibit inferior TF properties since it possessed the lowest strength and the largest grain size among the three materials according to Equation (6). Furthermore, MH should exhibit superior TF properties because it possessed the highest strength and the smallest grain size. In reality, it was correct for LH (except at the early stage of TF tests), but not correct for MH. In fact, it is HH that exhibited superior TF properties during the remaining TF test cycles. LH possessed the shortest crack length after 250 cycles of tests and should be largely attributed to the largest portion of coarse laths in the material. Whereas, the retained austenite, prior austenite size, and strength have little significance on crack propagation at the early stage of TF tests. Obviously, the high KAM value in MH and HH should be the main reason for the high crack propagation rate at the early stage of TF tests, since materials with high KAM values means high stored energy [36] and may facilitate the propagation of cracks from the surface [37]. After 500 cycles, the crack lengths of LH, MH, and HH were in descending order. LH exhibited the poorest TF resistance and should be attributed to the larger prior austenite and lower hardness value compared to the other two materials. The improvement of TF resistance properties of MH should be attributed to the increased hardness and the decreased grain size. Since the retained austenite only functions at the early stage of thermal treatment, further improvement of TF resistance in HH should be mainly attributed to the increased coarse laths (that is, the decreased KAM value). A few increases in prior austenite size did not deteriorate TF resistance properties in HH.

In conclusion, the area fraction of austenite has little effect on the TF properties of chromium hot-work steels processed by SLM. The enhanced TF properties should be attributed to the increased strength (or hardness) in the material and the refined grain size. Coarse laths (or the decreased KAMs value) also play an important role in the improvement of the TF properties in the material.

## 5. Conclusions

The effects of the content of alloying elements on the microstructure, phase compositions, high-temperature tensile strength, and TF properties of the SLM-processed chromium hot-work steels were investigated. The major findings are summarized as follows:

(1)  All the as-built SLM-processed chromium hot-work steels exhibited the martensite and retained austenite structures. Retained austenite content increased with the increase of the alloying additions.

(2)  The average martensitic blocks size was approximately 2.56 µm, 1.43 µm, and 1.72 µm for LH, MH, and HH, respectively. That is because the increase of V led to grain refinement in MH, whereas the increase of Cr and Mo led to some grain coarsening in HH. The content of coarse laths and the KAM value increased with the increase in martensitic block sizes.

(3)  LH, MH, and HH possessed the UTS of $1218 \pm 18$ MPa, $1281 \pm 19$ MPa, and $1258 \pm 22$ MPa, and hardness of $528 \pm 8$ HV$_5$, $597 \pm 12$ HV$_5$, and $592 \pm 9$ HV$_5$, respectively. All the materials experienced a secondary hardening process during TF tests, with the hardness of $573 \pm 9$ HV$_5$, $688 \pm 11$ HV$_5$, and $675 \pm 10$ HV$_5$, respectively, after TF tests.

(4)  LH exhibited the longest crack length after TF tests, and HH exhibited the shortest. Grain refinement led to the increase in strength and enhancement of TF resistance in MH. Although the grain coarsening and strength decrease may facilitate the crack propagation in HH, the increase in the amount of coarse laths enhanced the TF resistance in the material. It is suggested that the grain size and the amount of coarse laths are the most important factors to determine the TF resistance of chromium hot-work steels that are processed by SLM.

**Author Contributions:** M.W. contributed to the conception, the experiment of the study and wrote the manuscript; B.Y. helped perform the analysis with constructive discussions; Y.W. helped with grammar checking and helped perform the analysis with constructive discussions; X.G. helped perform the microstructural analysis with constructive discussions; B.L. helped perform the analysis with constructive discussions; W.L. helped perform the analysis with constructive discussions; Q.W. funded the article and helped perform the analysis with constructive discussions. All authors have read and agreed to the published version of the manuscript.

**Funding:** This work was supported by the Key Research and Development Project of Hubei Province, China (grant No. 2020BAB049); the Ronggui Street to Support Strategic Emerging Industry Development Project; the Wuhan Enterprise Technology Innovation Project (grant No. 2020010602012037); the Heilongjiang Provincial Natural Science Foundation of China (LH2020E089); the China Postdoctoral Science Foundation (2021M691970); and the Postdoctoral Innovation Project of Shandong Province (202103051).

**Institutional Review Board Statement:** Not applicable.

**Informed Consent Statement:** Not applicable.

**Data Availability Statement:** The processed data required to reproduce these findings are available to download from [https://data.mendeley.com/datasets/kbwpgytdtg/draft?a=6fa42a37-4c77-4218-a797-f6d45618e8aa], accessed on 3 April 2022.

**Acknowledgments:** The authors would also like to thank the State key Laboratory of Materials Processing and Die & Mould Technology, and the Analysis and Testing Center of HUST.

**Conflicts of Interest:** The authors declare no conflict of interest.

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
