# Peer review of "Effect of Cr, Mo, and V Elements on the Microstructure and Thermal Fatigue Properties of the Chromium Hot-Work Steels Processed by Selective Laser Melting"

_metals, doi:10.3390/met12050735_

Round 1

Reviewer 1 Report

The mandatory English language check of the manuscript has to be made.

Author Response

Reviewer 1

The mandatory English language check of the manuscript has to be made.

Thank you for helping us on the language. We have checked through the manuscript and all the revisions can be found by using the Track Changes function of MS Word.

Reviewer 2 Report

The paper submitted to metals treats about the influence of SLM parameters on the properties, microstructure and phase content  of hot work tool steel manufactured additively.

The idea of the paper is clear, the subject Is well chosen. The paper is written with the use of (as far as I can judge)  proper English. The reader can understand well everything and follow easily what the authors have In mind.  

The abstract is a little bit complicated as it defines many acronyms and by that I think authors should consider changing that a bit or move defining the acronyms to introduction.

The introduction Is well written, with a proper definition of the subject as well as proper choice of literature. I only miss a small part devoted to the properties and microstructure of this steel manufactured by DED methods (like lend or LMD). This can show differences and may be very useful in the discussion part.

The experimental procedure is written very well. In my opinion, its missing only one important parameter which is the size of the beam and power distribution in beam (gauusian?). The choice of the investigation methods is also very well done.

The description of the data is very professional and the authors give only justified statements and observations. The graphical presentation is very clear and well done. Sometimes I would suggest improving the size of the fonts to improve readability. Also I suggest adding measurement uncertainties in case of tensile properties and hardness. Otherwise, I think there is really not much to improve.

I must say as a final statement that for 64 reviews I have ever made for MDPI journals this work is second best. I strongly suggest for publication in almost this form or after little tuning following the abovementioned suggestions.

Author Response

Answer:

Thank you for all your wonderful comments and suggestions.

For the abstract section, acronyms were decreased and only the SLM, LH, MH, and HH were remained.

For the introduction section, as you said the properties and microstructure of this steel manufactured by SLM are very useful in the discussion part. We added this part in the third passage of Introduction section.

About the SLM process, as you suggested we added “The focused laser beam diameter is of 80 μm with a Gaussian distribution of power density.”

As you suggested that the measurement uncertainties in case of tensile properties and hardness should be add, we added the standard deviation for them in the manuscript (section 3.5, section 3.6, and conclusion).

Reviewer 3 Report

The manuscript entitled: 'Effect of Cr, Mo, and V elements on the microstructure and thermal fatigue properties of the chromium hot-work steels processed by selective laser melting' deals with the effect of carbide forming elements (Cr, Mo, and V) on the microstructures and TF properties of the chromium hot-work steel fabricated by SLM was systematically investigated. The TF resistance mechanism of SLM-processed H13 was discussed and revealed. However, I have the following minor concerns with the present manuscript.

  • All the SLM processing parameters should be introduced including hatch style/distance, etc.
  • Scale Bars are missing in Fig. 2
  • Typos in the manuscript need attention.

Author Response

Thank you for your help to point out these errors. We have revised these errors according you suggestions (section 2.2 and Figure 2). Besides, We have checked through the manuscript about the language and all the revisions can be found by using the Track Changes function of MS Word.
